# Comparison of Multiple Strategies for Precision Transgene Knock-In in *Gallus gallus* Genome via Microhomology-Mediated End Joining

**DOI:** 10.3390/ijms242115731

**Published:** 2023-10-29

**Authors:** Lijuan Wang, Jiaxin Sun, Zhipeng Liu, Qiang Zheng, Guojun Wang

**Affiliations:** The State Key Laboratory of Reproductive Regulation and Breeding of Grassland Livestock, College of Life Sciences, Inner Mongolia University, Hohhot 010070, China; 22008036@mail.imu.edu.cn (L.W.); 32008165@mail.imu.edu.cn (J.S.); 32108035@mail.imu.edu.cn (Z.L.); 32008114@mail.imu.edu.cn (Q.Z.)

**Keywords:** MMEJ, *GAPDH*, DF-1 cell line, precise integration, CRISPR/Cas9, safe harbor loci

## Abstract

Precision exogenous gene knock-in is an attractive field for transgenic *Gallus gallus* (chicken) generation. In this article, we constructed multiple Precise Integration into Target Chromosome (PITCh) plasmid systems mediated by microhomology-mediated end-joining (MMEJ) for large-fragment integration in DF-1 cells and further assess the possibility of *GAPDH* (glyceraldehyde-3-phosphate dehydrogenase) as a genomic safe harbor for chickens. We designed three targeted sgRNAs for the all-in-one plasmid at the 3′UTR of *GAPDH* near the stop codon. The donor-plasmid-carrying microhomology arms correspond to sgRNA and EGFP fragments in the forward and reverse directions. MMEJ-mediated EGFP insertion can be efficiently expressed in DF-1 cells. Moreover, the differences between the forward and reverse fragments indicated that promoter interference does affect the transfection efficiency of plasmids and cell proliferation. The comparison of the 20 bp and 40 bp microhomology arms declared that the short one has higher knock-in efficiency. Even though all three different transgene insertion sites in *GAPDH* could be used to integrate the foreign gene, we noticed that the G2-20R-EGFP cell reduced the expression of *GAPDH*, and the G3-20R-EGFP cell exhibited significant growth retardation. Taken together, G1, located at the 3′UTR of *GAPDH* on the outer side of the last base of the terminator, can be a candidate genomic safe harbor (GSH) loci for the chicken genome. In addition, deleted-in-azoospermia-like (*DAZL*) and actin beta (*ACTB*) site-specific gene knock-in indicated that MMEJ has broad applicability and high-precision knock-in efficiency for genetically engineered chickens.

## 1. Introduction

The chicken is one of our important protein sources and a valuable model for studying immunology and developmental biology [1]. Traditional genetic improvements based on beneficial natural genetic variation exist within the population [2]. With the wide application of Clustered Regularly Interspaced Short Palindromic Repeats (CRISPR) and CRISPR-associated nuclease 9 (Cas9), novel genetic variation and exogenous traits from other genera and species can be introduced into poultry [3,4]. With the application of critical technologies, such as the in vitro culture of PGCs, chicken genetic breeding is becoming increasingly attractive [5]. Many of the applications for enhancing disease resistance and producing biomedical materials rely on functional gene integration [6]. Previously, the generation of transgenic chicken was achieved by nonspecific insertions using retroviral vectors, phi-31 integrase, piggyBac, and Tol2 transposon [7,8,9]. This would always lead to gene silencing, cancer-related gene insertion, unexpected tissue-specific expression, frameshift in an open reading frame, and other genetic instabilities due to random integration [10,11,12]. However, site-specific gene integration, introducing an exogenous gene at an expected position in the genome without random or multi-copy insertions, provides a valuable path for chicken breeding.

As the most advanced gene-editing tool, CRISPR/Cas9 enables genome engineering by introducing double-strand breaks (DSBs) at specific genomic loci [13]. Classically, DSBs recruit endogenous repair machinery for either non-homologous end-joining (NHEJ) or homology-directed repair (HDR) to mediate genome editing [14]. NHEJ is a rapid, high-capacity pathway that joins two DNA ends with minimal reference to the DNA sequence [15]; it can occur in dividing and non-dividing cells. NHEJ can join widely separated DNA ends, resulting in large deletions, which can be widely used for gene knock-out. However, NHEJ is an error-prone repair pathway when joining DNA ends with or without small insertions and deletions, which can cause a frameshift in an open reading frame [14]. HDR is mainly used to generate gene knock-in and a precise repair pathway and is active during the late S/G2 phases (dividing cells). To accomplish this, HDR requires extensive sequence homology between the broken and donor DNA molecules and entails templated DNA synthesis as a critical step in the repair process [15]. HDR requires the specific genomic loci to be flanked by homology arms of approximately 800–6000 bp [16]. Isogenic DNA is needed, since mismatches in the homology regions cannot be tolerated [17,18]. Obviously, for most non-inbred livestock, this is challenging to achieve. Independently from NHEJ and HDR, MMEJ is an alternative DSB repair pathway known to be active during the M–early S phases when HDR is inactive [13,19]. The evidence suggests that when NHEJ or HDR is deficient, MMEJ is a robust and efficient alternative repair option [20,21]. Generally, MMEJ is utilized in short deletions of the intervening sequence, when short microhomologies exist in the two DNA ends [13]. As the extremely short homologous sequence (5–40 bp) for DSB repair and precise gene knock-in, MMEJ can be a suitable candidate tool for site-specific insertion in the chicken genome [13,22,23].

In order to stably and reliably express the newly integrated DNA in the interested tissues, the transgene should be precisely inserted into the GSH [11,13]. For humans and mammals, three intragenic sites (ROSA26, CCR5, and AAVS1) are recognized as GSHs for exogenous gene targeting [6,11,24,25,26,27,28,29,30,31]. In recent years, several genes served as the GSH to host transgenes in chicken cells. The chicken ovalbumin (*OV*) locus was proved to be useful for producing human interferon-β (*hIFN-β*) and human epidermal growth factor (*hEGF*) [32,33,34,35,36,37]. However, ovalbumin is a tissue-specific gene unsuitable for exogenous genes requiring constitutive expression. It was reported that endogenous avian virus (*EAV-HP*) can be a candidate safe harbor for constitutive expression. However, *EAV-HP* has multiple copies in the chicken genome, and its safety needs further assessment [38,39]. Previously, an EGFP cassette was precisely integrated into a *GAPDH* locus by HDR in DF-1 cells [40]. However, the initial targeting rate was very low (1.8%). Currently, two novel genomic safe harbor loci, *cHIPP* and *cROSA*, are found in the genome of DF-1 cells [41]. Nonetheless, further in vivo investigation is necessary for genetically engineered chickens.

Although the chicken was the first livestock to be sequenced, the generation of genetically modified chickens has yet to catch up to the generation of genetically modified mammals. Developing an efficient site-specific integration strategy in the chicken genome and verifying an ideal safe harbor are urgently needed. This article demonstrates that MMEJ mediates efficient foreign gene integration into the chicken genome. We compare multiple strategies for large-fragment integration in chicken *GAPDH* to optimize insertion efficiency. The results prove that the reverse knock-in of the foreign gene with a shorter homologous arm could promote the knock-in rate. In addition, the G1 site of *GAPDH* could be a GSH site for the chicken genome.

## 2. Results

### 2.1. MMEJ-Mediated Efficient Foreign Gene Integration into DF-1 Cell

As mismatches between the homologous arm of the primer and the target sequence significantly decrease the knock-in efficiency, we sequenced the target gene, *GAPDH*, in DF-1 cells (Appendix A). Then, we designed sgRNA based on the target gene sequence, located at the 3′UTR of *GAPDH* on the outer side of the last base of the terminator, named G1, as shown in Figure 1A and Appendix A. The PITCh plasmid system comprises an integrated Cas9 plasmid with a dual sgRNA (all-in-one plasmid) and another plasmid carrying the homologous arm and EGFP fragments. There are 20 bp homologous arms on both sides of the plasmid that match the DNA sequences on both sides of the sgRNA cleavage site. After 72 h of transfection, fluorescence-activated cell sorting (FACS) analysis showed that the proportion of EGFP-positive cells was 25.6% (Figure 1B,D,E), much higher than the HDR-mediated knock-in efficiency [40].

After puromycin selection, PCR and subsequent analysis were performed using three primer pairs (Appendix A). The PCR identification results showed that the entire length of the knock-in fragment was consistent with the design (Figure 1C). The sequencing analysis showed that the EGFP was precisely integrated into the Cas9 cutting target. Neither the deletion nor insertion mutation was detected (Figure 1F). In summary, we efficiently knocked foreign genes into the target of DF-1 endogenous genes.

### 2.2. Reverse Knock-In More Productively Performs Than Forward Insertion

Adding promoters to inserted fragments is a common strategy to enhance transgene expression. However, promoter interference occurs when the two promoters are relatively close [22]. We constructed another donor vector, an EF-1α promoter-driven EGFP-2A-Puro cassette with a polyA signal sequence, which was reversely added against the *GAPDH* gene to avoid promoter interference. The plasmid structure is shown in Figure 2A and Appendix A. After 72 h of transfection, the FACS results showed that the EGFP-positive cell rates were 25.6% and 32.1% in forward and reverse insertion cells, respectively (Figure 2B,C). After puromycin selection and cell amplification, the PCR and sequencing results showed that the reverse-inserted EGFP was accurately integrated into the target site, and no mutations were detected (Figure 2D). The Western blot further confirmed the expression of EGFP both in reverse- and forward-insertion cells (Figure 2E). As expected, the expression of *GAPDH* did not change.

Cell proliferation was assessed by colorimetric thiazolyl blue (MTT) assays and growth curves to verify the impact on cellular function. We found that the forward-inserted G1-20F-EGFP cells exhibited growth retardation. By contrast, the reverse-inserted G1-20R-EGFP cells exhibited no change in cell growth compared to the wild type (Figure 2F–H). This indicates that promoter interference affects the transfection efficiency of plasmids and cell proliferation, which was rarely noticed in genetically modified chicken research.

### 2.3. Short Homologous Arms Are More Efficient in Large-Fragment Integration

From the perspective of gene knock-in in HDR, large-segment knock-in always requires a longer homologous arm. Therefore, we constructed 40 bp forward and reverse insertion vectors, as shown in Figure 3A. The flow cytometry analysis showed that the transfection efficiency of plasmids containing 20 bp homologous arms was significantly higher than that of plasmids containing 40 bp homologous arms in both the forward and reverse directions. When inserted in the forward direction, the positive rates of EGFP cells in the 20 bp and 40 bp homologous arms were 25.6% and 19.2%, respectively (Figure 3B,C). When inserted in the reverse direction, they were 32.1% and 28.2%, respectively (Figure 3B,C). These results indicated that the short homologous arm had higher knock-in efficiency when the exogenous fragment was up to 2800 bp.

### 2.4. Different Transgene Insertion Sites Caused Diverse Effects on Integration Efficiency, Gene Expression, and Cell Viability

To explore the effects of the knock-in site on MMEJ efficiency, we selected two different sgRNAs in the 3′UTR before and after the *GAPDH* terminator, G2 and G3, respectively (Figure 4A). The T7E1 restriction endonuclease analysis showed that the cleavage efficiency among the three loci was 22.6%, 23%, and 20.6% (Figure 4B) [42]. We constructed three PITCh-directed donor vectors containing a 2.8 kb EGFP-2A-Puro-polyA reverse cassette flanked by 20 bp left and right microhomologies corresponding to the G1, G2, and G3 loci. The FACS results showed that the EGFP-positive cell rates of the reverse insertion vectors at the G1, G2, and G3 loci were 32.2%, 30.5%, and 33.3%, respectively. The knock-in efficiency of G1 and G3 was higher than that of G2 (Figure 4C–E).

The PCR analysis and sequencing results of the three stable transfected cells showed that EGFP could be accurately tapped into the three sites, and no mutations were detected (Figure 4F,G). The Western blot results confirmed the undifferentiated expression of EGFP at the protein level. However, it is worth noting that *GAPDH* expression in G2-20R-EGFP cells was significantly lower than that of the G1, G3, and wild-type cells (Figure 4H). In addition to that, we found that the cells of G2-20R-EGFP and G3-20R-EGFP exhibited more significant growth retardation than those of G1 (Figure 4I–K). By contrast, G1-20R-EGFP showed no significant difference compared to the wild type (Figure 4I,J). These results indicated that the insertion of foreign genes at the G2 and G3 sites before the stop codon can affect the expression of *GAPDH* and cell viability.

### 2.5. Broad Applicability of MMEJ-Assisted Targeting Vector for Gene Knock-In in the Chicken Genome

To further validate the broad applicability of the dual plasmid system, we selected another common housekeeping gene, *ACTB*, and a gene specifically expressed in chicken testicular tissue, *DAZL*, as candidates. After sequencing around the insertion site, two sgRNAs were designed for each (Figure 5A and Appendix A). Through T7E1 restriction endonuclease analysis (Figure 5B), we selected ACTB-sgRNA6 and DAZL-sgRNA36 for subsequent analysis and accordingly modified the homologous arm of the knock-in plasmid. The FACS results showed that the EGFP-positive cell rates of ACTB-sgRNA6 and DAZL-sgRNA36 were 17.3% and 13.8%, respectively (Figure 5C–E). Compared with the three sgRNAs of *GAPDH*, the different efficiencies indicated that MMEJ-assisted gene knock-in using CRISPR-Cas9 differed when different sgRNAs were designed for the same or different genes. PCR analysis and sequencing confirmed the absence of cells without modification (Figure 5F,G). This indicated that the MMEJ-assisted gene knock-in plasmid system demonstrated the broad applicability and high efficiency of precise knock-in in the chicken genome.

## 3. Discussion

In this article, we constructed multiple PITCh plasmid systems mediated by MMEJ for large-fragment integration in DF-1 cells. Our results proved that MMEJ mediated efficient foreign gene integration into chicken cells. Notably, short homologous arms were more efficient in large-fragment integration. Moreover, reverse knock-in performed more productively than forward insertion. By comparing three different transgene insertion sites in *GAPDH*, we noticed that they showed diverse effects on integration efficiency, gene expression, and cell viability. In addition, *DAZL* and *ACTB* site-specific gene knock-in indicated that this plasmid system had broad applicability and high precision knock-in efficiency.

The precise insertion of exogenous genes is essential for breeding and biological research, but more research is needed in chickens. In 2019, Ekaterina Antonova et al. designed sgRNA for the 3′UTR of the *GAPDH* gene and performed precise typing of EGFP using homology-directed repair (HDR) [40]. However, the vector contained a 999 bp left homologous arm and a 3093 bp right homologous arm, and the EGFP-positive cell rate was only 0.5%. Low integration efficiency often leads to difficulties in integrating PGCs with limited proliferation in the future. Therefore, there is an urgent need to improve the transfection efficiency of knock-in vectors. Herein, we reported the precise foreign gene knock-in technologies mediated by the CRISPR/Cas9 system with the cassette for MMEJ in chicken DF-1 cells. Since MMEJ can occur in most of the cell cycle, the knock-in frequency was up to 21.6%, suggesting the superiority of this strategy to HDR. In addition to that, the lengths of the microhomologies on the knock-in vector were just 20 bp, which a single PCR can easily add without amplification from genomic DNA.

Previous research about the distribution of endogenous retroviruses in the mammals and avian genome revealed that most of these fragments are enriched outside transcription units in reverse orientation relative to the host [43,44,45]. Considering the research of endogenous retrovirus biology, integration outside transcription units may be primarily benign and was preserved during evolution [11,46]. Our results indicated that promoter interference does exist at this knock-in site, and reverse insertion can make it easier for foreign genes to be integrated into the target site.

In previous studies targeting the *GAPDH* gene, no insertion of foreign genes was detected by sgRNAs before and after the terminator mediated by HDR [40]. Here, we successfully achieved the insertion at the G2 and G3 loci. Even though the sgRNA of G2 is in the 3′UTR region of *GAPDH*, it was found that knocking in at the G2 locus still had a significant impact on the expression of internal genes, which was not reported before. Since the G3-20R-EGFP cells showed growth retardation, we checked the potential off-target sites using the ChopChop and CRISPR Design Tool websites. We found 3 off-target sites in G1, 3 off-target sites in G2, and 17 off-target sites in G3. The high number of off-target sites in G3 may explain the abnormal growth outcomes of G3-20R-EGFP cells. These results indicated that G1 could be a candidate GSH for foreign gene integration in the chicken genome. In the future, further work is needed both in PGCs and in chickens.

## 4. Materials and Methods

### 4.1. Construction of PITCh Plasmids

pCRIS-PITChv2-GAPDH, the donor vector (Plasmid no.63672, Addgene, Watertown, MA, USA) containing the EGFP-2A-Puro cDNA cassette for knock-in, was constructed using two separate PCRs and In-Fusion HD cloning kit (Takara, Beijing, China). pX330A-GAPDH/PITCh, an all-in-one CRISPR/Cas9 vector for targeting the genomic *GAPDH* locus and the donor vector, was constructed as previously described using pX330A-1×2 (Plasmid no.58766, Addgene, Watertown, MA, USA) and pX330S-2-PITCh (Plasmid no.63670, Addgene, Watertown, MA, USA) vector [5]. The all-in-one and donor vector targeting *ACTB* and *DAZL* construct is the same as *GAPDH.* The oligonucleotides used for the sgRNA template are listed in Appendix A.

### 4.2. Cell Culture and Transfection

DF-1 cells were cultured in DMEM supplemented with 10% fetal bovine serum (Gibco, Waltham, MA, USA) at 37 °C, 5% CO_2_. The DF-1 cell line was obtained from National Collection of Authenticated Cell Cultures (Shanghai, China). Lipofectamine LTX (Life Technologies, Waltham, MA, USA) was used to transfect the plasmids following the supplier’s protocols. The numbers of plasmid concentrations, cell numbers, and plates used were as follows: for the T7E1 assay, 3.3 µg all-in-one CRISPR/Cas9 vectors into 4 × 10^5^ DF-1 cells using a six-well plate; for the FACS analysis and fluorescence observation of EGFP knock-in at the *GAPDH* locus, 5 µg of plasmids in total (the ratio of all-in-one CRISPR/Cas9 and PITCh donor vector is 2:1) into 4 × 10^5^ DF-1 cells using a six-well plate; for the genomic PCR, RNA extraction, Western blot, 13.75 µg of plasmids in total (the ratio of all-in-one CRISPR/Cas9 and PITCh donor vector is 2:1) into 2.2 × 10^6^ DF-1 cells using a 100 mm dish. After transfection, cells were cultured in the growth medium for 72 h and then selected with 1.5 μg/mL puromycin for 7 days.

### 4.3. T7E1 Assay

DF-1 cells were cultured and transfected as described above. After incubation at 37 °C, 5% CO_2_ for 72 h, cells were harvested, and genomic DNA was isolated using the DNeasy Blood and Tissue Kit (QIAGEN, Hilden, Germany). PCR to amplify endogenous loci was performed by PrimerSTAR GXL DNA Polymerase (Takara, Beijing, China) using the primers listed in Appendix A. The PCR products were purified by TIANgel Midi Purification Kit (TIANGEN, Beijing, China) according to the manufacturer’s instructions. A T7E1 assay to detect genetic alterations was performed under the manufacturer’s guidelines (M0302, NEB, Ipswich, MA, USA). Then, 200 ng of purified PCR products was denatured in NEBuffer 2 at 95 °C for 5 min and reannealed at a controlled rate of −2 °C/s for 95 °C −85 °C and −0.1 °C/s for 85 °C −25 °C, with a 4 °C hold. Samples were divided in half, and 1 µL of T7 Endonuclease Ⅰ or ddH2O was added, followed by incubation at 37 °C for 15 min. The T7E1 reaction was stopped with 0.25 M EDTA. Digestion products were analyzed by gel electrophoresis, and indel frequencies were calculated by densitometry in Image J v1.53e (NIH) using the formula: 100 × (1 − (1 − (b + c)/(a + b + c))^1/2^).

### 4.4. Genotyping and Sequencing of Knock-In Junctions

Genomic DNA was extracted using a DNeasy Blood and Tissue Kit (QIAGEN, Hilden, Germany) from the knock-in cells collected after puromycin selection. Gnomic PCR was performed using KOD FX Neo (Toyobo, Osaka, Japan) or PrimerSTAR GXL DNA Polymerase (Takara, Beijing, China). PCR products were gel-purified and Sanger sequenced. Primers used for the genotyping and sequencing are listed in Appendix A.

### 4.5. FACS Analysis

The cells were collected at 72 h post-transfection, suspended in FACS Buffer (DPBS supplemented with 0.1% BSA), and filtered with a Cell-Strainer Tube (FALCON, Tamaulipas, Mexico). The number of cells with 488-nm Laser (EGFP) and the corresponding fluorescence filters (Alexa Fluor^®^ 594). Then, 10,000 cells were recorded for each sample after the preliminary FSC/SSC gating. Fluorescence was analyzed using a Beckman Coulter CytoFLEX LX with FlowJo software v10.

### 4.6. Western Blot Analysis

The proteins were obtained from the extracts of transfected cells with the RIPA buffer (final concentrations: 125 mM Tris-HCl, pH 6.8, 4% SDS, 20% glycerol, 10% β-mercaptoethanol, and 0.005% bromophenol blue). The cell lysates were denatured at 95 °C for 10 min and separated by 10–20% SDS-polyacrylamide gels. Samples were transferred onto polyvinylidene fluoride membranes. After blocking with 5% skim milk, each sample was incubated with primary antibodies against EGFP (1:20,000, Proteintech, Wuhan, China), GAPDH (1:20,000, ABclonal, Wuhan, China), and β-actin (1:20,000, Proteintech, Wuhan, China) in TBS-T containing 5% skim milk at 4 °C overnight. The membranes were washed and then incubated with secondary antibodies for HPRT (1:2000, Abcom, Waltham, MA, USA). Signals were visualized with Luminata Forte Western HRP substrate (Millipore, Darmstadt, Germany) and raised using E-Blot 14.1.221220.1 (E-Blot, Shanghai, China).

### 4.7. Off-Target Analysis of sgRNA

The potential off-target sites in the chicken genome for selected sgRNAs were identified online using the ChopChop tool “https://chopchop.cbu.uib.no/ (accessed on 15 March 2023)” and CRISPR Design Tool “https://crispr.mit.edu (accessed on 15 March 2023)” websites.

### 4.8. Growth Curve

EGFP^+^ cell lines were plated on a 24-well plate at 4 × 10^4^ cells/well. Cells were collected at 24, 48, 72, 96, and 120 h. Cell numbers were determined using an automated cell counter (Countess Ⅱ FL, Invitrogen, Waltham, MA, USA). Experiments were performed for three replicates.

### 4.9. Determination of Cell Proliferation

Cell proliferation was assessed by colorimetric thiazolyl blue (MTT) assay. Stably transfected cells were plated on a 96-well plate at 6 × 10^3^ cells/well. After an incubation period of 24 h at 37 °C, the tetrazolium salt MTT (CT02, Sigma-Aldrich, St. Louis, MO, USA) was added to a final concentration of 0.5 mg/mL, and incubation was continued for 4 h. Add 100 µL of the solubilization solution into each well and incubate overnight. Metabolization of MTT was quantitated by measuring the absorbance at 550 nm using a microplate reader (SpectraMax Mini, Molecular Devices, Shanghai, China). Experiments were performed for five replicates.

### 4.10. Statistical Analyses

Statistical tests were performed using GraphPad Prism 8. Two-tailed paired two-sample *t*-tests, and *t*-tests after one-way analysis of variance (ANOVA), were used to compare the WT with KI cells. Significant statistical differences are noted as * *p* < 0.05, ** *p* < 0.01, *** *p* < 0.001, and **** *p* < 0.0001. The data are presented as the mean ± S.D.

## 5. Conclusions

In conclusion, we demonstrated that the MMEJ-assisted gene knock-in could successfully target *GAPDH*, *ACTB*, and *DAZL* in the chicken genome. We evaluated the performance of three strategies, including the promoter integration direction, the length of microhomology arms, and the different transgene insertion sites in *GAPDH*. These results confirmed that the short homologous arm performs a higher knock-in efficiency, and the reverse-inserted foreign gene cassette exhibits an advantage in cell viability over the forward-inserted foreign gene cassette. In addition, a CRISPR/Cas9 target gene sequence located at the 3′UTR of *GAPDH* on the outer side of the last base of the terminator, named G1, could be a GSH site for the chicken genome. Furthermore, *DAZL* and *ACTB* site-specific gene knock-in indicated that MMEJ has broad applicability and high precision regarding knock-in efficiency. This study provided a practical approach to the breeding of genetically engineered chickens.

## Figures and Tables

**Figure 1 ijms-24-15731-f001:**
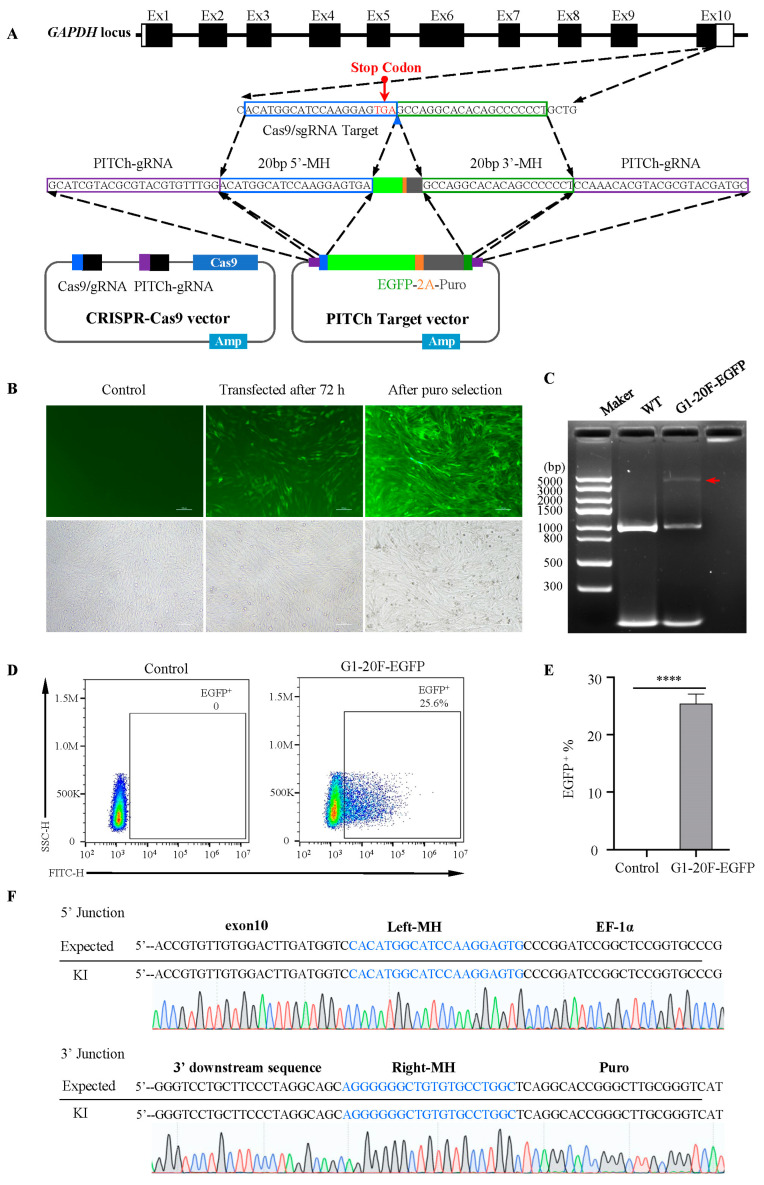
MMEJ-mediated gene knock-in at *GAPDH* locus in DF-1 cells. (**A**) A simplified schematic of PITCh-mediated EGFP knock-in into the 3′UTR of *GAPDH*. The CRISPR-Cas9 vector targeting near the stop codon at the 10th exon *GAPDH* is designed. The donor vector contains an EGFP cDNA connected to an EF-1α promoter for 3′UTR fusion. (**B**) DF-1 cells are transfected after 72 h, and stably transfected cells after puromycin selection are viewed under a fluorescence microscope. Scale bar: 100 µm. (**C**) PCR of the site-specific insertion in G1-20F-EGFP cell lines. The red arrow indicates the full length of the knock-in fragment. (**D**,**E**) Flow cytometry analysis of EGFP-positive cells 72 h after transfection. **** *p* < 0.0001; all values are presented as mean ± S.D. *T*-test was used to compare the WT with KI cells. (**F**) Sequencing analysis of stably transfected G1-20F-EGFP cell lines.

**Figure 2 ijms-24-15731-f002:**
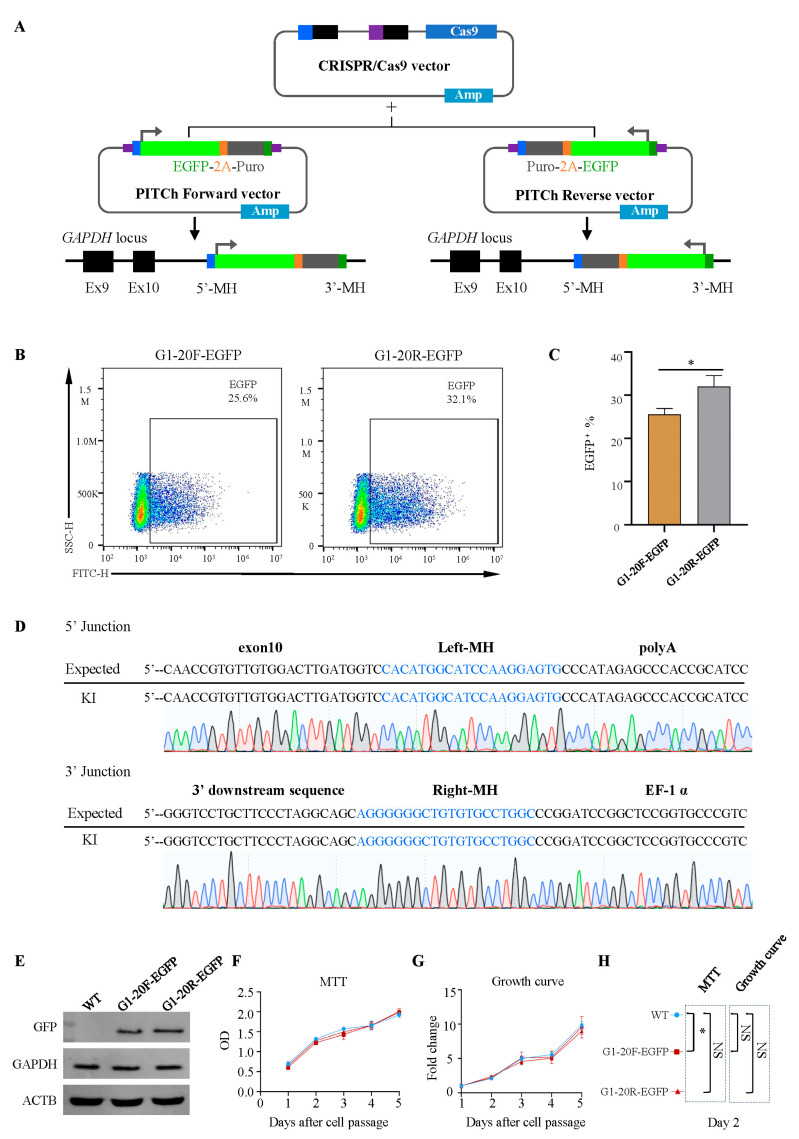
Comparisons between the forward and reverse insertions. (**A**) Construction of an all-in-one CRISPR-Cas9 vector harboring Cas9 nuclease and two gRNAs, a *GAPDH* target gRNA, and a generic PITCh-gRNA (left). Construction of the donor vector expressing EGFP linked forward and reverse (right). (**B**,**C**) Flow cytometry analysis of G1-20F-EGFP- and G1-20R-EGFP-positive cells 72 h after transfection. * *p* = 0.0175; all values are presented as mean ± S.D. *T*-test was used to compare. (**D**) Sequencing analysis of stably transfected G1-20R-EGFP cell lines. (**E**) Western blots of EGFP protein, *GAPDH*, and *ACTB* for Control, G1-20F-EGFP, and G1-20R-EGFP cell lines. (**F**–**H**) MTT assay, * *p* < 0.05; NS, not significant. *T*-tests were used after one-way ANOVA.

**Figure 3 ijms-24-15731-f003:**
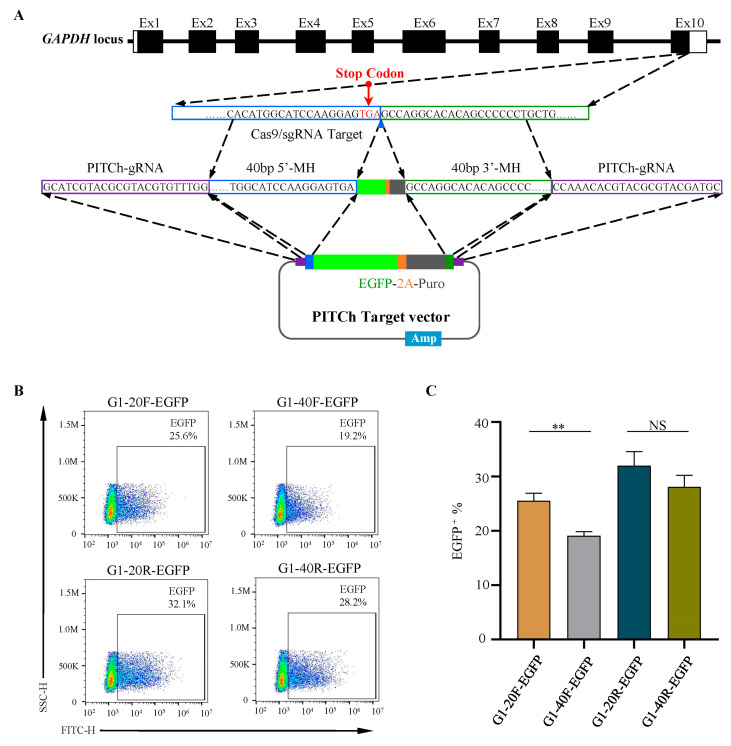
Comparisons between short and long homologous arms. (**A**) A schematic illustration of donor vector construction harboring 40 bp microhomologies. (**B**,**C**) Flow cytometry analysis of EGFP-positive cells 72 h after transfection for different PITCh vectors. ** *p* = 0.0019; NS, not significant. All values are presented as mean ± S.D. *T*-test was used after one-way ANOVA.

**Figure 4 ijms-24-15731-f004:**
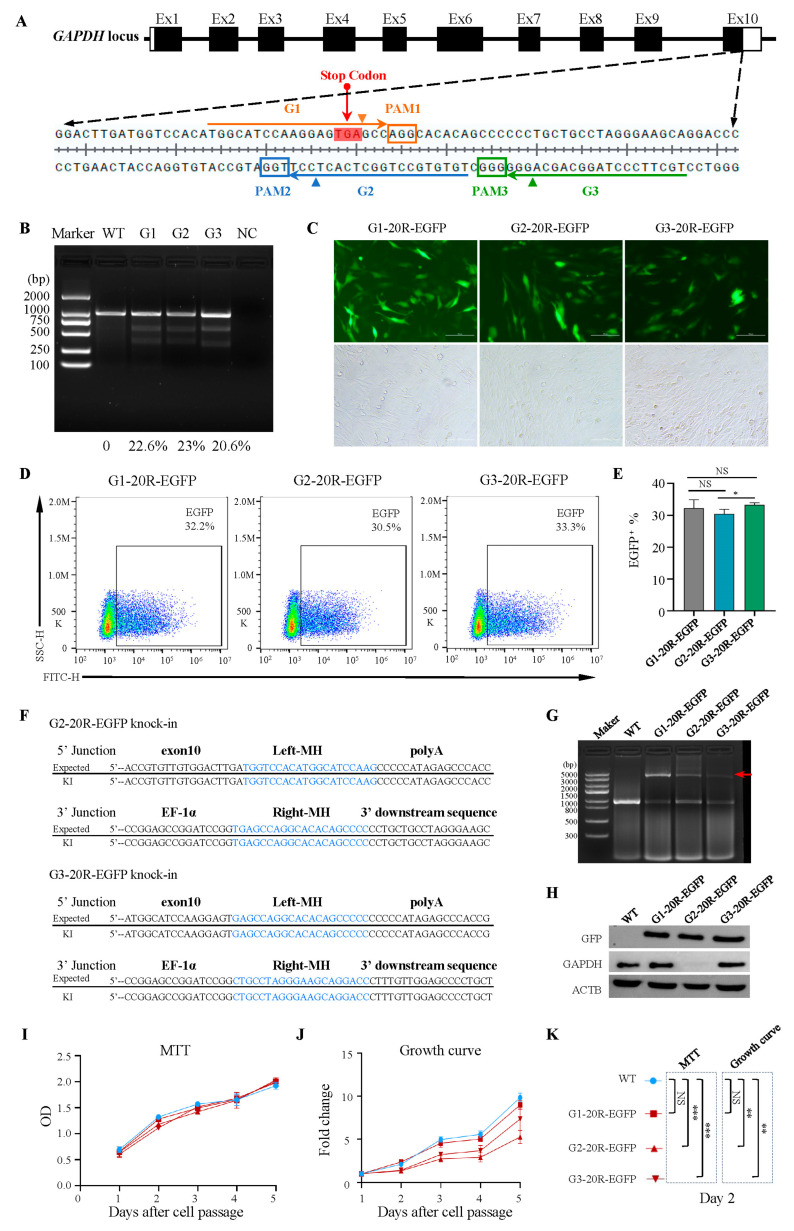
Insertion in different target sites caused diverse effects. (**A**) A schematic illustration of three locus-specific gRNAs targeting *GAPDH*: G1, G2, and G3. (**B**) T7 endonuclease Ⅰ assay of the *GAPDH* gene mutation in DF-1 cells. (**C**) DF-1 cells were transfected after 72 h under a fluorescence microscope. Scale bar: 100 µm. (**D**,**E**) Flow cytometry analysis of EGFP-positive cells 72 h after transfection for G1-20R-EGFP, G2-20R-EGFP, and G3-20R-EGFP. * *p* = 0.0334; NS, not significant. All values are presented as mean ± S.D. *T*-test was used after one-way ANOVA. (**F**) Sequencing analysis of stably transfected G2-20R-EGFP and G3-20R-EGFP cell lines. (**G**) PCR of the site-specific insertion in stably transfected EGFP cell lines. The red arrow indicates the full length of the knock-in fragment. (**H**) Western blots of EGFP protein, *GAPDH*, and *ACTB* for Control, G1-20R-EGFP, G2-20R-EGFP, and G3-20R-EGFP cell lines. (**I**–**K**) MTT assay, from right to left; NS, not significant; *** *p* = 0.001, *** *p* = 0.006. Growth curve, from left to right; NS, not significant; ** *p* = 0.0012, ** *p* = 0.0017. All values are presented as mean ± S.D. *T*-tests were used after one-way ANOVA.

**Figure 5 ijms-24-15731-f005:**
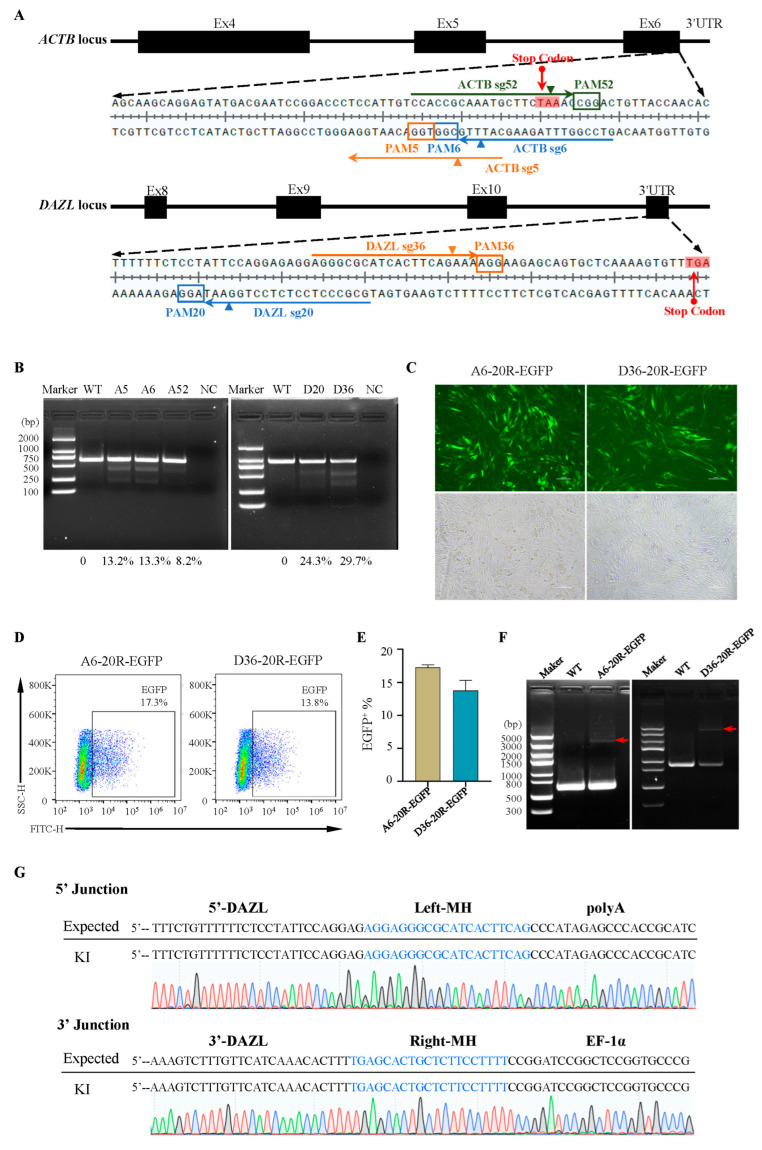
MMEJ-mediated gene knock-in at *ACTB* and *DAZL* loci in DF-1 cells. (**A**) sgRNA design for gene integration at *ACTB* and *DAZL* loci. (**B**) T7 endonuclease Ⅰ assay of the mutation in DF-1 cells. (**C**) DF-1 cells were transfected by targeted vectors after 72 h under a fluorescence microscope. Scale bar: 100 µm. (**D**,**E**) Flow cytometry analysis of EGFP-positive cells 72 h after transfection for A6-20R-EGFP and D36-20R-EGFP. (**F**) PCR of the site-specific insertion in stably transfected cell lines. The red arrow indicates the full length of the knock-in fragment (*ACTB*: 3476 bp and *DAZL:* 3729 bp). (**G**) Sequencing analysis of stably transfected DAZL36-EGFP cell lines.

## Data Availability

Data are contained within the article.

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
