# Peer review of "Comparison of Multiple Strategies for Precision Transgene Knock-In in *Gallus gallus* Genome via Microhomology-Mediated End Joining"

_ijms, 2023, doi:10.3390/ijms242115731_

Round 1

Reviewer 1 Report

This paper demonstrated that the MMEJ-assisted gene knock-in could successfully target the endogenous gene in the chicken genome. The results are substantial and useful for further study. 

Author Response

Thank you very much for taking the time to review this manuscript. Please see the attachment.

Reviewer 2 Report

This is an excellent submission.  The paper is well written.  The methodology is appropriate.  The results are exciting, significant and have important relevance. My only concern is the quality of the figures.  The size and colors of the fonts are not easily readable.  I would greatly encourage the authors to re-draft their figures.

Author Response

Thank you very much for taking the time to review this manuscript. As requested, we are submitting a revised version of our manuscript (ijms-2608359) titled “Comparison of multiple strategies for precision transgene knock-in in Gallus gallus Genome via microhomology-mediated end joining” by Lijuan Wang, Jiaxin Sun, Zhipeng Liu, Qiang Zheng, and Guojun Wang. We would like to thank the reviewers for their valuable comments and efforts to improve the manuscript. Please find the detailed responses below and the corrections highlighted in yellow in the re-submitted files.

Comments 1: [My only concern is the quality of the figures. The size and colors of the fonts are not easily readable. I would greatly encourage the authors to re-draft their figures.]

Response 1: Thank you for pointing this out. We agree with this comment. Therefore, we have [re-drafted and improved the resolution of all figures in our manuscript. We have enlarged the font in Figures 1 to 5, simplified the colors in Figures 1A, 2A, and 3A, and improved the resolution of Figures 1A, 1B, 1C, 2F, 2G, 3A, 3C, 4A, 4C, 4E, 4G, 4I, 4J, 5A, 5C, 5E, and 5F. The new figures can be found on Page 4, Line 112; Page 6, Line 142; Page 8, Line 163; Page 10, Line 189; Page 12, Line 218.]

Reviewer 3 Report

The authors show the construction of multiple Precise Integration into Target Chromosome (PITCh) plasmid systems mediated by microhomology-mediated end-joining (MMEJ) for large fragments integration in DF-1 cells. They assess the possibility of the GAPDH gene as a genomic safe harbor for chicken. Three different transgene insertion sites in GAPDH deemed useful to integrate the foreign gene, but some differences were observed.  Furthermore, DAZL and ACTB site-specific gene knock-in indicated that MMEJ has a broad applicability and a high precision in regard to knock-in efficiency for genetically engineered chicken.

The methods applied and results are well written and clearly presented.

Resolution of figures has to be improved.

Minor typos have to be amended.

Keywords have to be checked with the title to avoid doubling of words.

The advantages and problems and indications for the NHEJ, HDR and MMEJ should be discussed in more detail.

Conclusions have to be improved.

Line 341: "the endogenous gene in the chicken genome" please clarfy, not clear.

LIne 341 "Comparisons of these strategies indicate that" here not clear what you are comparing

Line 344 "Besides, the G1 site of GAPDH could be a GSH site for the chicken genome." not clear

Please amend the conclusion section and set a focus of the outcome of your experiments. Also, emphasize your achievements in a much better way.

No comments

Author Response

Thank you very much for taking the time to review this manuscript. As requested, we are submitting a revised version of our manuscript (ijms-2608359) titled “Comparison of multiple strategies for precision transgene knock-in in Gallus gallus Genome via microhomology-mediated end joining” by Lijuan Wang, Jiaxin Sun, Zhipeng Liu, Qiang Zheng, and Guojun Wang. We would like to thank the reviewers for their valuable comments and efforts to improve the manuscript. Please find the detailed responses below and the corrections highlighted in yellow in the re-submitted files.

Comments 1: [Resolution of figures has to be improved.]

Response 1: Thanks for your suggestion. Agree. Therefore, we have [re-drafted and improved the resolution of all figures in our manuscript. We have enlarged the font in Figures 1 to 5, simplified the colors in Figures 1A, 2A, and 3A, and improved the resolution of Figures 1A, 1B, 1C, 2F, 2G, 3A, 3C, 4A, 4C, 4E, 4G, 4I, 4J, 5A, 5C, 5E, and 5F. The new figures can be found on Page 4, Line 112; Page 6, Line 142; Page 8, Line 163; Page 10, Line 189; Page 12, Line 218.]

Comments 2: [Minor typos have to be amended.]

Response 2: Agree. Thank you for your reminder. We are sorry for our careless mistake. We have modified [the number from “3.4” to “2.4” on Page 8, Line 168, and from “3.5” to “2.5” on Page 11, Line 202. We have capitalized Zhipeng Liu’s initials on Page 16, Lines 365-367. We have made some minor editing of English language in introduction (Page 2, Paragraph 2, Line 55, and 61), and conclusions (Page15, Paragraph 6, Lines 354-355) highlighted in yellow in re-submitted manuscript.]

Comments 3: Keywords have to be checked with the title to avoid doubling of words.

Response 3: Thank you for pointing this out. We agree with this comment. Therefore, we have modified [the Keyworks form “MMEJ; DF-1 cell line; knock-in; CRISPR/Cas9; chicken genome” to “MMEJ; GAPDH; DF-1 cell line; precise integration; CRISPR/Cas9; safe harbor loci” on Page 1, Line 27.]

Comments 4: [The advantages and problems and indications for the NHEJ, HDR and MMEJ should be discussed in more detail.]

Response 4: Agree. We have rewritten the paragraph and added [details about the advantages, problems, and indications for NHEJ, HDR, and MMEJ on Page 2, Paragraph 2, Lines 49-52, Lines 56-58, Lines 64-66.]

Comments 5: [Conclusions have to be improved. Line 341: "the endogenous gene in the chicken genome" please clarify, not clear. Line 341 "Comparisons of these strategies indicate that" here not clear what you are comparing. Line 344 "Besides, the G1 site of GAPDH could be a GSH site for the chicken genome." not clear. Please amend the conclusion section and set a focus of the outcome of your experiments. Also, emphasize your achievements in a much better way.]

Response 5: Thank you for pointing this out. We agree with this comment. Therefore, we have rewritten the conclusion section according to the Reviewer’s suggestion. [Line 341“the endogenous gene in the chicken genome” was revised to “GAPDH, ACTB, and DAZL in the chicken genome”. Line 341, "Comparisons of these strategies indicate that" was modified to “We evaluated the performance of three strategies, including promoter integration direction, length of microhomology arms, and different transgene insertion sites in GAPDH”. Line 344, “Besides, the G1 site of GAPDH could be a GSH site for the chicken genome.” was changed to “a CRISPR/Cas9 target gene sequence located at the 3'UTR of GAPDH on the outer side of the last base of the terminator named G1” Page 15, Line 351-360.]

Reviewer 4 Report

The paper demonstrates generally high quality. The Abstract synthesised the main work methods and the main findings of the study. The Introduction provides necessary background for doing the research and the cited references are relevant. However, the aim of the study is rather vague. It should be formulated in a more clear manner.

The methodological approach is correct. The experimental work is performed on a very high level. The results are clearly presented and illustrated in adequate graphic material.  The discussion of the results is rather concise. In line 245-246 the authors refer to previous studies , without mentioning references. Almost all the references cited in discussion are about the human genome. Are there more references about avian genome or that of the other farm animals ? Should be better to cite.

Author Response

Thank you very much for taking the time to review this manuscript. As requested, we are submitting a revised version of our manuscript (ijms-2608359) titled “Comparison of multiple strategies for precision transgene knock-in in Gallus gallus Genome via microhomology-mediated end joining” by Lijuan Wang, Jiaxin Sun, Zhipeng Liu, Qiang Zheng, and Guojun Wang. We would like to thank the reviewers for their valuable comments and efforts to improve the manuscript. Please find the detailed responses below and the corrections highlighted in yellow in the re-submitted files.

Comments 1: [However, the aim of the study is rather vague. It should be formulated in a clearer manner.]

Response 1: Thank you for pointing this out. We agree with this comment. Therefore, we have added [“Although the chicken was the first livestock to be sequenced, the generation of genet-ically modified chickens has yet to catch up to mammals. Developing an efficient site-specific integration strategy in the chicken genome and verifying an ideal safe harbor is urgently needed.” (Page 2, Lines 84-87) to clarify the aim of the study.]

Comments 2: [On lines 245-246 the authors refer to previous studies, without mentioning references.

Response 2: Thanks for your suggestion. [We have added the reference 40 on Page 13, Line 256.]

Comments 3: [Almost all the references cited in discussion are about the human genome. Are there more references about avian genome or that of the other farm animals? Should be better to cite.]

Response 3: We sincerely appreciate the valuable comments. We agree with this comment.  [We have checked the literature carefully and added references 44 and 45 about avian and other animals into the discussion part in the revised manuscript (Page 13, Line 250). The expression “human” was revised to “mammals and avian” (Page 13, Lines 248-249)]
